# Soluble ST2 in Patients with Carotid Artery Stenosis—Association with Plaque Morphology and Long-Term Outcome

**DOI:** 10.3390/ijms24109007

**Published:** 2023-05-19

**Authors:** Stefan Stojkovic, Stephanie Kampf, Olesya Harkot, Maja Nackenhorst, Mira Brekalo, Kurt Huber, Christian Hengstenberg, Christoph Neumayer, Johann Wojta, Svitlana Demyanets

**Affiliations:** 1Department of Internal Medicine II, Division of Cardiology, Medical University of Vienna, 1090 Vienna, Austria; stefan.stojkovic@meduniwien.ac.at (S.S.);; 2Department of General Surgery, Division of Vascular Surgery, Medical University of Vienna, 1090 Vienna, Austria; 3Department of Pathology, Medical University of Vienna, 1090 Vienna, Austria; 43rd Medical Department with Cardiology and Intensive Care Medicine, Klinik Ottakring, 1160 Vienna, Austria; 5Medical School, Sigmund Freud University, 1020 Vienna, Austria; 6Ludwig Boltzmann Institute for Cardiovascular Research, 1090 Vienna, Austria; 7Core Facilities, Medical University of Vienna, 1090 Vienna, Austria; 8Department of Laboratory Medicine, Medical University of Vienna, 1090 Vienna, Austria

**Keywords:** carotid artery disease, soluble suppression of tumorigenesis 2 (sST2), prognostic biomarker

## Abstract

Interleukin (IL-33) and the ST2 receptor are implicated in the pathogenesis of atherosclerosis. Soluble ST2 (sST2), which negatively regulates IL-33 signaling, is an established biomarker in coronary artery disease and heart failure. Here we aimed to investigate the association of sST2 with carotid atherosclerotic plaque morphology, symptom presentation, and the prognostic value of sST2 in patients undergoing carotid endarterectomy. A total of 170 consecutive patients with high-grade asymptomatic or symptomatic carotid artery stenosis undergoing carotid endarterectomy were included in the study. The patients were followed up for 10 years, and the primary endpoint was defined as a composite of adverse cardiovascular events and cardiovascular mortality, with all-cause mortality as the secondary endpoint. The baseline sST2 showed no association with carotid plaque morphology assessed using carotid duplex ultrasound (B 0.051, 95% CI −0.145–0.248, *p* = 0.609), nor with modified histological AHA classification based on morphological description following surgery (B −0.032, 95% CI −0.194–0.130, *p* = 0.698). Furthermore, sST2 was not associated with baseline clinical symptoms (B −0.105, 95% CI −0.432–0.214, *p* = 0.517). On the other hand, sST2 was an independent predictor for long-term adverse cardiovascular events after adjustment for age, sex, and coronary artery disease (HR 1.4, 95% CI 1.0–2.4, *p* = 0.048), but not for all-cause mortality (HR 1.2, 95% CI 0.8–1.7, *p* = 0.301). Patients with high baseline sST2 levels had a significantly higher adverse cardiovascular event rate as compared to patients with lower sST2 (log-rank *p* < 0.001). Although IL-33 and ST2 play a role in the pathogenesis of atherosclerosis, sST2 is not associated with carotid plaque morphology. However, sST2 is an excellent prognostic marker for long-term adverse cardiovascular outcomes in patients with high-grade carotid artery stenosis.

## 1. Introduction

Atherosclerotic disease is the primary global contributor to morbidity and mortality, with carotid atherosclerosis playing a fundamental role in the occurrence of stroke due to occlusion of the carotid artery [1]. Despite the substantial benefit of modern revascularization strategies, the use of statins, and success in modulating risk factors, the long-term prognosis of patients with carotid atherosclerosis is still earmarked by a high risk of developing adverse vascular events and, subsequently, the need for improvement in the risk stratification modalities [2].

Suppression of tumorigenesis 2 (ST2) is a receptor for interleukin-33 (IL-33) and is known to be associated with the T-helper cells-2 response. ST2 is a member of the IL-1 receptor superfamily. ST2 exists in two isoforms: soluble ST2 (sST2) and transmembrane ST2 (ST2L). The IL-33/ST2 axis is involved in the pathogenesis of atherosclerosis [3,4]. The components of the IL-33/ST2 system are present in human carotid plaques [5]. IL-33 acts by binding to its transmembrane receptor, ST2L. The soluble form of ST2 (sST2) negatively regulates IL-33 signaling and inhibits the actions of IL-33 by its neutralization, thus preventing the binding of IL-33 to the transmembrane ST2L [6].

In addition, ST2, in its soluble form, is an inflammatory marker that has been shown to be associated with cardiovascular disease. Circulating sST2 has emerged as a valid prognostic biomarker not only in patients with myocardial infarction and heart failure (HF) but also in other chronic and acute conditions such as, e.g., acute aortic dissection or chronic obstructive pulmonary disease [7,8,9,10]. However, data on circulating sST2 in patients with carotid atherosclerosis are still rare and controversial. An earlier study did not find a predictive value of sST2 for future cardiovascular events during a three-year follow-up [11]. In contrast, another study detected sST2 as an independent prognostic determinant of symptomatic cerebrovascular disease and all-cause mortality during a five-year follow-up in patients who underwent carotid plaque endarterectomy [12].

As patients with carotid atherosclerosis are at high risk of cardiovascular events, the identification of circulating biomarkers for improved risk stratification is of special importance, especially for the long-term outcome. Therefore, we aimed to determine the possible predictive value of sST2 in a cohort of patients with symptomatic and asymptomatic carotid atherosclerosis during a 10-year follow-up. Additionally, we looked at the possible association of sST2 with carotid plaque morphology as well as carotid artery stenosis-related symptoms.

## 2. Results

The baseline clinical characteristics of the patient population are given in Table 1. Overall, the median age was 69 years, and 68% of patients were male. Furthermore, 35% of patients presented with symptomatic carotid stenosis. The most frequent comorbidities were hypertension (89%), followed by hypercholesterolemia (52%), peripheral (40%), and coronary (35%) artery disease. Symptomatic patients had higher baseline LDL and lower HDL as compared to asymptomatic patients. Symptomatic patients also had a lower grade of stenosis and more often echolucent and mixed plaques in carotid ultrasound, whereas asymptomatic patients had more often echogenic, i.e., calcified plaques (Table 1).

The baseline levels of circulating sST2 were similar in asymptomatic and symptomatic patients (10. 9 ng/mL, IQR 6.4–17.8 vs. 11.0 ng/mL, IQR 5.7–17.7, *p* = 0.965, Table 1). In logistic regression analysis, sST2 was not associated with symptomatic carotid artery stenosis (B = −0.105, 95% CI −0.432–0.214, *p* = 0.517). There was also no association of sST2 with carotid plaque morphology as determined by carotid ultrasound (B = 0.051, 95% CI −0.145–0.248, *p* = 0.609) and histological AHA classification (B = −0.032, 95% CI −0.194–0.130, *p* = 0.698). As depicted in Figure 1A, patients with echogenic, echolucent, and mixed plaques had similar sST2 concentrations (*p* = 0.810 across groups). There was also no difference in sST2 between groups according to histological AHA classification (Figure 1B, *p* = 0.248 across groups). In the multivariable logistic regression model, age (B = 0.023, 95% CI 0.001–0.045, *p* = 0.038), male sex (B = 6.8, 95% CI 3.1–10.4, *p* < 0.001), and coronary artery disease (CAD, B = 7.6, 95% CI 2.3–12.9, *p* = 0.005) were significantly correlated with baseline sST2 levels, while there was a tendency for peripheral artery disease (PAD, B = 3.1, 95% CI −0.3–6.5, *p* = 0.073); however, it did not reach statistical significance.

The patients were followed up for 10 years, with a median follow-up time of 6.7 years (IQR 3.1–8.0). The follow-up data were available for 167 patients (98.2%), and 3 patients were lost to follow-up. A total of 59 patients (34.7%) reached the composite primary endpoint, and 58 (34.1%) patients died during the follow-up. Of these, 28 patients died of cardiovascular causes.

The Cox proportional hazard regression models were assessed to investigate the predictive value of circulating sST2 for the composite primary endpoint. Circulating sST2 predicted adverse cardiovascular outcomes in a univariate Cox proportional hazard regression model with HR 2.4 (95% CI 1.3–4.4, *p* = 0.006, Table 2). Similarly, sST2 predicted all-cause mortality following carotid endarterectomy with HR 2.4 (95% CI 1.3–4.5, *p* = 0.008, Table 2). In a multivariable Cox proportional hazard regression model, sST2 independently predicted the primary endpoint after adjustment for age, sex, CAD, and PAD (HR 1.8, 95% CI 1.05–3.3, *p* = 0.034, Table 2). In contrast, the association of sST2 with all-cause mortality was lost in the same multivariable model (HR 1.8, 95% CI 0.9–3.5, *p* = 0.081, Table 2), with age being the most significant predictor of all-cause mortality (HR 1.04, 95% CI 1.008–1.073, *p* = 0.013).

In the overall cohort, the baseline median sST2 level was 11 ng/mL (IQR 6.1–17.8). Forty-two patients (24.7%) had sST2 in the highest quartile, above 17.8 ng/mL. Patients with high sST2 levels (>17.8 ng/mL) had a significantly higher event rate for the primary endpoint as compared to patients with lower sST2 (log-rank *p* < 0.001, Figure 2A). Similarly, high baseline sST2 was also associated with a high rate of all-cause mortality (log-rank *p* = 0.035, Figure 2B).

In the exploratory analysis of individual outcome endpoints, sST2 significantly predicted cardiovascular mortality (log-rank *p* = 0.013, Figure 3A), the progression of CAD and myocardial infarction (log-rank *p* = 0.004, Figure 3B), as well as the progression of PAD (log-rank *p* = 0.012, Figure 3C). In contrast, baseline sST2 did not predict future TIA/stroke, or carotid artery restenosis (log-rank *p* = 0.193, Figure 3D).

## 3. Discussion

In the present study, we could show that circulating sST2 is not associated with carotid plaque morphology or symptomatic carotid artery stenosis. Baseline sST2 levels could, however, independently predict long-term adverse cardiovascular events in patients with high-grade carotid artery stenosis undergoing carotid endarterectomy. The predictive value of sST2 was mainly driven by cardiovascular mortality, progression of CAD, myocardial infarction, and PAD. In contrast, sST2 did not predict the progression of carotid artery disease or the future rate of stroke or TIA.

The understanding of the role of the IL-33/ST2L/sST2 system in atherosclerosis underwent a paradigm shift since the description of IL-33 as a new member of the IL-1 superfamily in 2005 [3,4]. Although described in a murine atherosclerosis model as an atheroprotective cytokine [13], extracellular IL-33 induces proinflammatory, prothrombotic, and proangiogenic activation of human endothelial cells [5,14,15] and stimulates the release of procoagulant microvesicles from human monocytes [16]. All these processes are known to be implicated in the development and progression of atherosclerotic lesions [17]. Therefore, the protective or harmful effects of IL-33 in vascular pathologies are thought to be dependent on cellular and temporal expression as well as on experimental conditions [3].

The relevance of sST2 as a diagnostic and prognostic biomarker was investigated in different vascular pathologies such as CAD and acute myocardial infarction [7,8,18,19], stroke [20], PAD [21], acute aortic dissection [22], and abdominal aortic aneurysms (AAA) [23]. Higher levels of sST2 were shown to be an excellent predictive marker for poor prognosis, especially in acute settings such as myocardial infarction [7,8], but also in stable CAD [18]. On the other hand, sST2 was not able to predict the progression of AAA and was not associated with ischemic outcomes following infrainguinal angioplasty with stent implantation in PAD, as was shown by our group previously [21,23]. A recent meta-analysis showed that sST2 levels did not differ significantly between patients with ischemic heart disease or myocardial infarction and healthy individuals [24]. However, currently, there are no standardized methods for the quantification of circulating sST2 levels; therefore, the comparison of the concentrations measured in different studies with different ELISA kits is not reliable. Furthermore, studies in the meta-analysis included very heterogeneous patient populations with HFpEF, HFrEF, coronary artery disease, microvascular angina, acute myocardial infarction, etc., making a comparison between studies even more difficult. However, one finding was consistent in all studies: patients with myocardial infarction and heart failure and high sST2 values always have a worse outcome as compared to patients with lower sST2.

Patients with type 2 diabetes and carotid artery stenosis have an increased risk of early in-stent restenosis after carotid artery stenting [25]. In the present study, one-third of the patients had type 2 diabetes. We observed no differences in carotid artery restenosis rate following carotid endarterectomy in patients with and without type 2 diabetes.

The association of circulating sST2 with morphological plaque characteristics and symptoms, as well as the prognostic relevance of sST2 in carotid atherosclerosis, is still controversial [11,12,26]. In the present study, we could neither show an association of sST2 with carotid plaque morphology nor with symptom presentation. This is in line with previously published results from the Athero-Express cohort [11]. The same study showed, however, no association of sST2 with secondary cardiovascular events in patients who underwent carotid endarterectomy and were subsequently followed for 3 years. In contrast, we could show here that after long-term follow-up of up to 10 years, sST2 is an excellent and independent predictor for adverse cardiovascular events and cardiovascular mortality in patients with high-grade carotid artery stenosis. A higher event rate (34.7% vs. 20%), as well as longer follow-up (median 6.7 years vs. 3 years), could possibly explain these conflicting results regarding the outcome between the present study and the Athero-Express cohort [11]. In fact, careful analysis of the Kaplan–Meier survival and failure plots of our study shows a separation of the curves after 3 and more years of follow-up. Thus, patients with carotid artery disease undergoing carotid endarterectomy and elevated sST2 levels prior to surgery have a high long-term risk for adverse cardiovascular events.

A recent study by Scicchitano et al., showed a predictive value of sST2 for all-cause mortality during a 5-year follow-up in a smaller cohort of eighty-two patients who underwent carotid plaque endarterectomy [12]. In our study, sST2 also predicted all-cause mortality in univariate Cox regression analysis; however, this association was lost after adjustment for age, sex, CAD, and PAD in the multivariable model, with both age and male sex as independent predictors of all-cause mortality. In the study of Scicchitano et al., 83.3% of non-survivors were female, compared to 19% in our cohort [12]. Distinct sex distribution could explain these apparently conflicting results between the two cohorts, as sex-related differences in circulating sST2 were previously demonstrated, with significantly higher sST2 in male patients [27]. Sex-related differences in carotid atherosclerosis are known, with plaque features such as plaque size, composition, and morphology more common or larger in men compared to women [28]. In the present study, age, male sex, and CAD were significantly correlated with baseline sST2 levels in a multivariable logistic regression model. Higher sST2 levels in males compared with females were observed in participants of the Framingham Heart Study [27]. Although male patients have a higher sST2 than female patients [27], sST2 seems to be a better predictor of all-cause mortality in female patients than in male patients [12]. These results further emphasize the importance of sex in the risk stratification of patients with cardiovascular disease.

It could be speculated that different human cells and tissues, including vascular endothelial cells, the heart, and adipose tissue, are responsible for circulating sST2 levels. The fraction of sST2 produced by each particular tissue would most probably relate to the presence or absence of pathological stimuli known to trigger the secretion of sST2 [4]. Our group demonstrated previously that human vascular and cardiac cells produce different amounts of sST2 protein, with endothelial cells seeming to be the main source of sST2 in the cardiovascular system [29]. We previously described ST2L as being present in human carotid atherosclerotic plaques [5]. A recent study demonstrated that IL-33 and ST2 expression were significantly higher in vulnerable plaques and significantly correlated with the degree of inflammatory cell infiltration in these plaques [26]. This is in line with our previous results, which demonstrate IL-33 and ST2 expression in the core of the plaque in regions with high tissue factor and urokinase expression [14,15]. The differences in ST2L expression in atherosclerotic lesions seem to be cell-dependent, as similar ST2L expression in atherosclerotic plaques of asymptomatic and symptomatic patients in T cells and endothelial cells of neo-angiogenic vessels was found, but more ST2L in macrophages of symptomatic patients was described by another study [30]. Furthermore, ST2 is also expressed in monocytes, and IL-33 induces monocytes to shift towards a pro-inflammatory and pro-thrombotic phenotype [16]. In spite of clear correlations between IL-33 and the ST2 receptor with plaque inflammation and vulnerability, circulating sST2 does not mirror plaque ST2 concentration or plaque morphology.

### Limitations

The small sample size increases the possibility of type II error and represents one of the study’s limitations. However, the high event rate and long-term follow-up of 10 years with only 3 patients lost to follow-up should overcome statistical concerns. The study was powered for a cumulative primary endpoint, and individual cardiovascular endpoints should be viewed as exploratory and hypothesis-generating only. Furthermore, blood samples for assessment of sST2 were taken at a single time point, namely at baseline one day prior to surgery. Thus, further insight into disease progression or response to secondary prevention therapy based on possible dynamic changes of sST2 over time cannot be gained from the present study. Finally, the currently available ELISA cannot distinguish between free and complexed sST2 [31].

## 4. Materials and Methods

### 4.1. Patient Population

A total of 170 consecutive patients with carotid artery stenosis undergoing carotid endarterectomy were included in the study. All surgical procedures were performed at the Department of Vascular Surgery, Medical University of Vienna, Austria. The indication for surgery included symptomatic carotid artery stenosis or high-grade asymptomatic stenosis (>70%). The degree of luminal narrowing was determined by carotid duplex ultrasound and/or computed tomography angiography using the criteria of the North American Symptomatic Carotid Endarterectomy Trial [32]. Patients were considered symptomatic if they had experienced a stroke, transient ischemic attack, or amaurosis fugax ipsilateral to the carotid artery stenosis within 6 months before carotid endarterectomy. Patients with hemorrhagic, lacunar, or cardioembolic strokes were excluded from the study. Lacunar strokes were defined as the presence of characteristic primary motor, primary sensory, or sensory-motor symptoms in combination with deep white-matter lesions or basal ganglia lesions 1 cm or less in diameter. Suspected cardioembolic stroke in patients with atrial fibrillation, recent myocardial infarction, unstable angina, recent congestive heart failure, and valvular disease prone to the production of emboli was reviewed by an independent, experienced cardiologist.

A 10-year follow-up was collected for all patients, and the primary cardiovascular endpoint was defined as the composite of cardiovascular death, myocardial infarction, transient ischemic attacks (TIA), or stroke, as well as atherosclerosis progression in the coronary or peripheral arteries requiring either interventional (percutaneous coronary intervention or peripheral balloon angioplasty with and without stenting) or surgical revascularization (aortocoronary bypass or peripheral bypass). The secondary endpoint was defined as all-cause mortality.

The study has been reviewed and approved by the Ethics Committee of the Medical University of Vienna, and all study subjects gave written informed consent (“MedUniWien Karotisendarterektomie Biobank”, ethics approval number 2449/2020).

### 4.2. Carotid Plaque Morphology

The morphological evaluation of carotid artery stenosis before surgery was assessed using carotid duplex ultrasound and/or computed tomography angiography. Plaques were classified according to echogenicity into echolucent (non-calcified), echogenic (calcified), and mixed plaques (partly calcified) in the expert vascular ultrasound laboratory at the Medical University of Vienna [33,34]. For further validation of the ultrasound characterization of the plaques, computed tomography angiography was applied to classify plaques into non-calcified, calcified, and partly calcified (mixed) plaques, and an agreement of 86% was achieved between the duplex ultrasound and computed tomography angiography characterization of the plaques. For histological classification, endarterectomy specimens were formalin-fixed and embedded in paraffin. Transverse sections were cut every 3 mm along the plaque by experienced technicians. Specimens were stained with hematoxylin-eosin and elastic-van-Gießen staining according to standard in-house protocol. Histologic specimens were classified according to a modified AHA classification based on the morphological description [35,36,37].

### 4.3. Soluble ST2 Measurement

Peripheral venous blood samples were collected into serum tubes containing a clotting activator (Greiner Bio One, Kremsmünster, Austria), held at room temperature for 1 h, centrifuged at 1000× *g* for 10 min, aliquoted, and stored at −80 °C until analyses. Levels of sST2 were quantified using the commercial human ST2/IL-1 R4 DuoSet^®^ ELISA Kit (R&D Systems, Minneapolis, MN, USA), according to the manufacturer’s instructions and as previously described by our group [9,21,23]. The ST2/IL-33R ELISA Kit has an intra-assay variability of 4.4–5.6% and an inter-assay variability of 5.4–7.1%. This ELISA has a sensitivity of 13.5 pg/mL and is specific for natural and recombinant human ST2, as well as free ST2 and IL-33 complexed ST2. The assay range of the human ST2/IL-33R ELISA is 31.3–2000 pg/mL.

### 4.4. Statistical Analysis

The categorical variables are summarized as counts and percentages and are compared by the χ^2^-test or Fisher’s exact test, as appropriate. However, the continuous variables are expressed as the median and interquartile range (IQR) and compared by the t-test or the Mann–Whitney U test in cases of non-normal distribution. Univariate and multivariate linear regression models were fit to evaluate the associations of sST2 with carotid plaque morphology and baseline clinical presentation (symptomatic/asymptomatic). The univariate Cox proportional hazard regression model was fit to assess whether sST2 could significantly predict long-term cardiovascular outcomes. Hazard ratios (HR) are given as HR per one standard deviation increase (HR per 1-SD). Kaplan-Meier survival plots were constructed in groups according to sST2 expression above or below the highest quartile value to compare the time-dependent discriminative power of circulating sST2. Two-sided *p*-values of 0.05 indicated statistical significance. SPSS 22.0 (IBM Corporation, Armonk, NY, USA) and STATA version 12 (StataCorp LLC, College Station, TX, USA) were used for all statistical analyses.

## 5. Conclusions

Although IL-33 and ST2 play a role in the pathogenesis of atherosclerosis, circulating sST2 is neither associated with carotid plaque morphology nor with symptomatic cerebrovascular events, i.e., TIA and stroke, in patients with high-grade carotid artery stenosis. However, circulating levels of sST2 are a promising prognostic marker for adverse cardiovascular events after long-term follow-up and might improve the risk stratification of patients with high-grade carotid artery stenosis undergoing carotid endarterectomy.

## Figures and Tables

**Figure 1 ijms-24-09007-f001:**
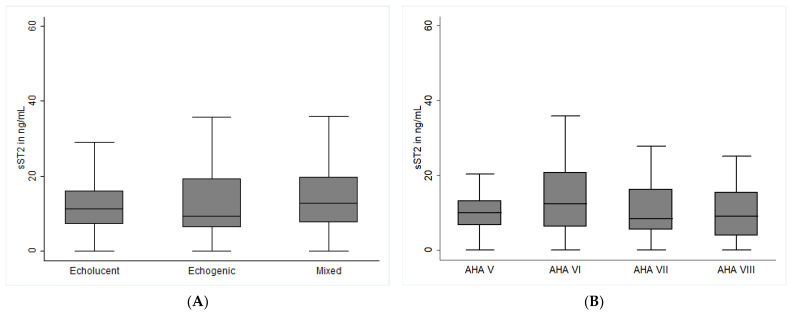
**Soluble ST2 expression and carotid plaque morphology in patients with carotid artery stenosis.** Baseline sST2 levels and carotid plaque morphology were determined as described in the methods section. (**A**) depicts sST2 levels in patients according to carotid ultrasound. Plaques were classified according to echogenicity into echolucent (non-calcified), echogenic (calcified), and mixed plaques (partly calcified), as described in the Methods section. (**B**) depicts sST2 levels in groups according to modified AHA classification: AHA type V fibroatheroma; AHA Type VI complex lesion with possible surface defect, hemorrhage, or thrombus; AHA type VII calcified lesion; AHA Type VIII fibrotic lesion without lipid core and with possible small calcifications Concentrations are depicted as medians with interquartile range and given in ng/mL. *p*-values < 0.05 were considered significant.

**Figure 2 ijms-24-09007-f002:**
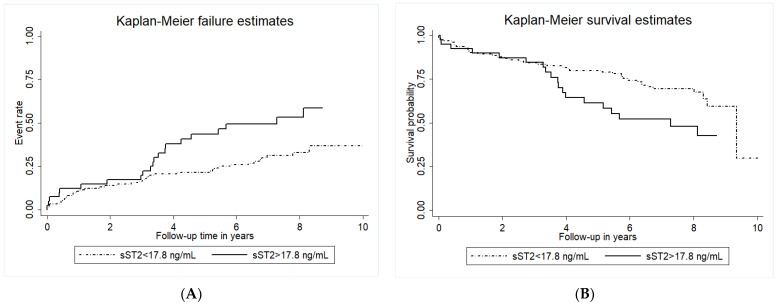
Cumulative incidence of adverse cardiovascular events and Kaplan-Maier survival estimates for all-cause mortality according to circulating sST2 levels. Kaplan-Meyer analyses for the cumulative incidence of adverse cardiovascular events (primary endpoint, panel (**A**)) and Kaplan-Maier survival estimates for all-cause mortality (panel (**B**)) according to baseline circulating sST2 levels. Circulating sST2 was measured as described in the methods section. The group with circulating sST2 above the cut-off of 17.8 ng/mL is indicated by the full line, and the dotted line indicates the group with sST2 below the cut-off.

**Figure 3 ijms-24-09007-f003:**
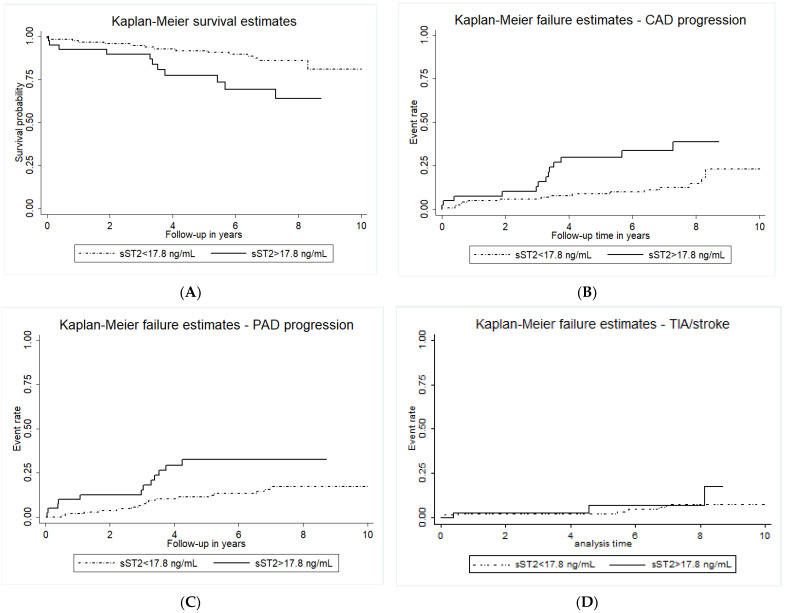
**Kaplan-Maier survival estimates for cardiovascular mortality and incidence of individual cardiovascular events.** Kaplan-Maier survival estimates for cardiovascular mortality (panel (**A**)), the incidence of CAD progression and acute myocardial infarction (panel (**B**)), PAD progression (panel (**C**)), as well as TIA and stroke (panel (**D**)) according to baseline circulating sST2 levels. Circulating sST2 was measured as described in the methods section. The group with circulating sST2 above the cut-off of 17.8 ng/mL is indicated by the full line, and the dotted line indicates the group with sST2 below the cut-off.

**Table 1 ijms-24-09007-t001:** Baseline clinical characteristics of the patient cohort.

	Overall (*n* = 170)	Asymptomatic (*n* = 111)	Symptomatic (*n* = 59)	*p*-Value
**Demographics**				
Age	69 (64–74)	69 (64–73)	69 (62–74)	0.888
Sex (male)	116 (68)	70 (63)	46 (78)	0.057
**Characteristics of carotid artery stenosis**				
Grade of stenosis	90 (85–95)	90 (85–95)	85 (80–90)	0.003
Stenosis grade ≥ 90%	104 (61)	79 (71)	25 (42)	<0.001
Contralateral stenosis	51 (30)	31 (28)	20 (34)	0.378
Plaque morphology				
Echogenic	57 (33)	45 (40)	12 (20)	0.005
Mixed	35 (21)	16 (14)	19 (32)	
Echolucent	63 (37)	39 (35)	24 (41)	
Histological AHA classification				
Type V fibroatheroma	32 (19)	21 (19)	11 (19)	0.770
Type VI complex lesion	72 (42)	47 (42)	25 (42)	
Type VII calcified lesion	27 (16)	16 (14)	11 (19)	
Type VIII fibrotic lesion	26 (15)	19 (17)	7 (12)	
**Comorbidities and risk factors**				
Transient ischemic attack	30 (18)	0 (0)	30 (51)	<0.001
Stroke	29 (17)	0 (0)	29 (49)	<0.001
History of stroke/TIA	32 (19)	17 (15.3)	15 (25.9)	0.074
Acute myocardial infarction	39 (23)	26 (23)	13 (22)	0.882
Coronary artery disease	59 (35)	36 (32)	23 (39)	0.350
Peripheral artery disease	69 (40)	50 (45)	19 (32)	0.123
Arterial hypertension	151 (89)	98 (88)	53 (90)	0.536
Hypercholesterolemia	88 (52)	56 (50)	32 (54)	0.560
Diabetes mellitus type 2	57 (33)	41 (37)	16 (27)	0.222
Adipositas (BMI > 30)	44 (26)	27 (24)	17 (29)	0.483
Smoking active	48 (28)	31 (28)	17 (29)	0.824
Past smoker	48 (28)	33 (30)	15 (25)	0.640
Pack-years	0 (0–40)	0 (0–38)	0 (0–40)	0.991
COPD	36 (21)	24 (22)	12 (20)	0.929
**Concomitant medication**				
Statin	154 (91)	101 (91)	53 (90)	0.939
Antiplatelet	162 (95)	106 (95)	56 (95)	0.691
ACE-inhibitor/ARB	98 (58)	68 (61)	30 (50)	0.347
**Laboratory parameters**				
Total cholesterol, mg/dL	163 (138–194)	163 (137–191)	156 (142–205)	0.877
LDL, mg/dL	84 (67–110)	77 (65–108)	98 (82–115)	0.018
HDL, mg/dL	50 (39–62)	54 (41–66)	40 (36–51)	0.001
Triglycerides, mg/dL	131 (93–199)	121 (91–189)	154 (94–209)	0.144
High-sensitivity CRP, mg/dL	0.5 (0.2–1.2)	0.5 (0.2–1.1)	0.5 (0.2–1.5)	0.655
Soluble ST2, ng/mL	10.9 (6.1–17.7)	10. 9 (6.4–17.8)	11.0 (5.7–17.7)	0.965

Continuous data are shown as median (interquartile range). Dichotomous data are shown as n (%). Mann–Whitney test was used for the statistical comparison of continuous variables between symptomatic and asymptomatic plaques and Fisher’s exact test for the categorical variables. Chi square test was used for categorical variables of more than two classes (ultrasound and histology). AHA, American Heart Association; TIA, transient ischemic attack; BMI, body mass index; ACE, angiotensin converting enzyme; ARB, angiotensin receptor blocker; CRP, C-reactive protein; HDL, high-density lipoprotein; LDL, low-density lipoprotein.

**Table 2 ijms-24-09007-t002:** Prognostic value of soluble ST2 for the composite primary endpoint and all-cause mortality in patients with carotid artery stenosis.

	HR Per 1-SD	95% CI	*p*-Value
**Univariable**			
Primary endpoint *	2.4	1.3–4.4	0.006
All-cause mortality	2.4	1.3–4.5	0.008
**Multivariable model**			
Primary endpoint *	1.8	1.05–3.3	0.034
All-cause mortality	1.8	0.9–3.5	0.081

* Primary endpoint was defined as the composite of cardiovascular death, myocardial infarction, transient ischemic attacks, or stroke, as well as atherosclerosis progression in the coronary or peripheral arteries requiring either interventional (percutaneous coronary intervention or peripheral balloon angioplasty with and without stenting) or surgical revascularization (aortocoronary bypass or peripheral bypass); Univariable and multivariable Cox proportional hazard regression models were fit to investigate the predictive value of circulating sST2 for the composite primary endpoint and all-cause mortality. Multivariable model was adjusted for age, sex, coronary and peripheral artery disease; HR per 1-SD, hazard ratio per one increase in standard deviation; CI, confidence interval.

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
