# Peer review of "Soluble ST2 in Patients with Carotid Artery Stenosis—Association with Plaque Morphology and Long-Term Outcome"

_ijms, 2023, doi:10.3390/ijms24109007_

Round 1

Reviewer 1 Report

I really liked the manuscript presented.

I was very interested to read a study on the search for a biomarker of carotid atherosclerotic plaque condition. It may seem that the article mainly presents negative results, however, I believe that publication of them is necessary for  the field of molecular biology of atherosclerosis.

I think that manuscript is relevant for the field of carotid artery disease. Of course manuscript is clear and presented in a well-structured manner.

I have no complaints about scientifically soundness and the experimental design, about ethics statements, methods and statistical analysis. I think that all the plots and tables are appropriate and easy to understand.

I very like the conclusions, as well as indications of limitations. Particularly interesting is the explanation of the conflict with the Athero-Express cohort, i.e. data on the long-term (3 and more years) risk of adverse events.

I see self-citation in at least 8 out of 34 literature references, however, most of them are not very recent (2011-2017) and this study can be considered as obtaining newer and more relevant data.

Also note that if the "Materials and Methods" section is number 4, this breaks the sequence of references in the text. However, this is a technical comment that does not affect the high evaluation of the study.
I hope that these interesting results will be published as soon as possible!

Reviewer 2 Report

An important work on atherosclerosis and its connection with inflammatory cells in the carotid arterial system. 

ST2 is an inflammatory marker that has been shown to be associated with cardiovascular disease. ST2 is a receptor of interleukin-33 (IL-33) that is associated with Th-2 and is a member of the IL-1 receptor family. ST2 has two forms: soluble ST2 (sST2) and transmembrane ST2. The form that has been most frequently investigated in cardiovascular diseases is sST2. By binding to IL-33, sST2 causes a decrease in levels of IL-33, which has cardioprotective effects. Therefore, in previous studies, sST2 has been shown to be associated with coronary artery disease, myocardial infarction, and ischemic heart disease but this being a definitive, strong connection is still far from being firmly established or affirmed.

- first, the authors may wish to clarify the mechanism of this inflammatory marker in a more easy-to-understand way (like I tried above), the reader finds it hard to understand from the Introduction what sST2 is. 

- second, the authors may lower the tone of the wording "The involvement of interleukin (IL)-33/suppression of tumorigenesis 2 (ST2) axis in 49 the pathogenesis of atherosclerosis is undoubted." as in fact reference 4 cited by them argues the prognostic role of L-33/ST2/sST2 signalling network and reference 3 is one of their own articles. They should rather change the word "undoubted" with "established" or "well-studied".

- in fact, it is increasingly appreciated that the pathophysiological importance of IL-33 is highly dependent on cellular and temporal expression. A recent meta-analysis showed that sST2 levels did not differ significantly between patients with IHD or MI and healthy individuals (https://doi.org/10.3389/fcvm.2021.697837). The authors may wish to reference and comment on this meta-analysis either in the Discussion or in the Introduction and clarify to the reader what is the latest position on these data.

- Abstract: clear

- Methods: your patients had a lot of AHA Complex lesion type, please state what "complex lesion" means from a pathological perspective

- Discussion: could sST2 be associated with more plaque "activation" and vulnerable plaques rather than general atherosclerosis? In your study there was no significant correlation with symptomatic carotid artery stenosis but there was a significantly higher event rate for major events. This could reflect that sST2 could reflect unstable plaques? Please comment on this. Something similar was discovered in coronaries: https://doi.org/10.1080/1354750X.2022.2032350

- many of your patients had other atherosclerotic diseases (CAD, PAD) and other major risk factors (HTN, dyslipidemia, DM). This harmful interplay may interact and drive major events. A better study of sST2 would be in this high-risk population? For example, DM can influence stent restenosis and cardiovascular events after carotid stenting - there are population at risk where biomarkers could weigh more than in other populations ( please cite DOI: 10.1155/2022/4196195)

Thank you.

English scientific sound and quality are good.

Reviewer 3 Report

Reviewer comments and suggestions

The study aimed to investigate the association of sST2 with carotid atherosclerotic plaque morphology, symptom presentation as well as prognostic value of sST2 in patients undergoing carotid endarterectomy. 

The authors included 170 consecutive patients with high-grade asymptomatic or symptomatic carotid artery stenosis undergoing carotid endarterectomy and patients were followed-up for 10 years. The result showed that sST2 was an independent predictor for long-term adverse cardiovascular events after adjustment for age, sex and coronary artery disease (HR 1.4, 95% CI 1.0 – 2.4, p=0.048), but not for all-cause mortality (HR 1.2, 95% CI 0.8 – 1.7, p=0.301). 

Patients with high baseline sST2 levels had significantly higher adverse cardiovascular event rate as compared to patients with lower sST2. In conclusion, the authors suggested that sST2 is an excellent prognostic marker for long-term adverse cardiovascular outcomes in patients with high-grade carotid artery stenosis.

Overall, the manuscript was well written. However, a few concerns/comments needed to be explained/modified. 

  1. Line 35, “sST2 is not associated with carotid plaque morphology” What does it mean, is it important to add at the conclusion part
  2. The cited references in the text differ according to the MDPI guidelines.
  3. Line 52 please explore the study
  4. Line 62-63 please add more points to the cited article
  5. Comments for Figure 1 it would be nice if the authors describe all mentioned in the x axis
  6. Comments for figure 3 how it was differentiated with the ROC curve that is also used to calculate the cut-off values.
  7. Line 189-191 Here the authors could add his/ her results.
  8. Line 213-214 is there any possible reason for the sex differences.
  9. Line 261-262 please mention the number.
  10. Please check the guidelines of mdpi, it seems that the authors need to modify all the references.

Round 2

Reviewer 2 Report

Excellent revision, the authors addressed all my comments in a point by point manner. The rebuttal letter was clear and the modifications brought in the main manuscript shaped the article into a much better form. I also noticed a considerable amount of research by the same team in the soluble ST2 field, I wish to thank them for their overall merits as well.